# *Drosophila* Contributions towards Understanding Neurofibromatosis 1

**DOI:** 10.3390/cells13080721

**Published:** 2024-04-21

**Authors:** Kalliopi Atsoniou, Eleni Giannopoulou, Eirini-Maria Georganta, Efthimios M. C. Skoulakis

**Affiliations:** 1Institute for Fundamental Biomedical Research, Biomedical Sciences Research Center “Alexander Fleming”, 16672 Athens, Greece; atsoniou@fleming.gr (K.A.); egiannopoulou@umh.es (E.G.); 2Laboratory of Experimental Physiology, Medical School, National and Kapodistrian University of Athens, 11527 Athens, Greece

**Keywords:** Neurofibromatosis 1, growth, behavioral deficits, animal models, *Drosophila*

## Abstract

Neurofibromatosis 1 (NF1) is a multisymptomatic disorder with highly variable presentations, which include short stature, susceptibility to formation of the characteristic benign tumors known as neurofibromas, intense freckling and skin discoloration, and cognitive deficits, which characterize most children with the condition. Attention deficits and Autism Spectrum manifestations augment the compromised learning presented by most patients, leading to behavioral problems and school failure, while fragmented sleep contributes to chronic fatigue and poor quality of life. Neurofibromin (Nf1) is present ubiquitously during human development and postnatally in most neuronal, oligodendrocyte, and Schwann cells. Evidence largely from animal models including *Drosophila* suggests that the symptomatic variability may reflect distinct cell-type-specific functions of the protein, which emerge upon its loss, or mutations affecting the different functional domains of the protein. This review summarizes the contributions of *Drosophila* in modeling multiple NF1 manifestations, addressing hypotheses regarding the cell-type-specific functions of the protein and exploring the molecular pathways affected upon loss of the highly conserved fly homolog dNf1. Collectively, work in this model not only has efficiently and expediently modelled multiple aspects of the condition and increased understanding of its behavioral manifestations, but also has led to pharmaceutical strategies towards their amelioration.

## 1. Introduction

### 1.1. Neurofibromin and Neurofibromatosis 1

Neurofibromatosis type 1 (NF1) (OMIM#162200) is an autosomal dominant multi-symptomatic disorder affecting ~1 in 2000–3000 individuals worldwide and is caused by inherited or de novo mutations in the *NF1* tumor suppressor gene (chromosome 17q11.2) [1]. The *NF1* gene encodes neurofibromin (Nf1) isoforms via ostensibly tissue-specific alternative splicing of the primary transcript [2]. Nf1 isoforms are large, ~2800 aa, multifunctional proteins, which are involved in the regulation of multiple signaling pathways, including Ras/MAPK, Raf/MEK/ERK, PI3K/AKT/mTOR, and cAMP/PKA [3,4,5,6]. It is generally unknown at the moment whether all or some isoforms are equally involved in regulating these diverse cascades, or whether there is differential, Nf1-isoform-specific regulation of particular signaling mechanisms. In accordance with this notion, an isoform containing additional amino acids in the domain acting as a negative regulator of Ras signaling (see below) is a less efficient regulator of this signaling cascade and differentially distributed in various tissues and neuronal subtypes [7]. The *NF1* gene is expressed ubiquitously during embryonic development with the adult pattern established after the first week of postnatal life [8,9,10]. In adults, it is present in most tissues, but expression is highest in neurons, glial cells, oligodendrocytes, astrocytes, leukocytes, and Schwann cells [9,11,12,13]. 

NF1 predominantly presents with benign tumors of the peripheral nervous system, called plexiform neurofibromas, as well as increased susceptibility to development of other brain and nerve tumors, including optic pathway gliomas and malignant peripheral nerve sheath tumors (MPNSTs) [14]. While typically considered an oncogenic predisposition syndrome, it is also associated with a number of non-tumor manifestations, including skeletal and skin pigmentation abnormalities (intense freckling, multiple Café-au-lait spots—CALSs), reduced overall growth, or short stature [14,15,16,17]. Neuronal impairments include executive and higher-order functions such as planning, visuospatial skills, reading/vocabulary, and motor coordination, sleep, learning, and cognitive impairments, including autism-like symptoms and attention deficits [18,19,20,21,22,23]. Like neurofibromas, cognitive dysfunction is among the hallmark symptoms of NF1. The prevalence of specific learning disabilities (SLDs) is notably elevated among individuals diagnosed with NF1. Approximately 20–60% of NF1 patients meet the criteria for SLD, which is significantly higher compared to its occurrence in the general population (5–15%) [24]. Although the cognitive symptoms do not progress with age [25], cognitive dysfunction leading to behavioral problems is perceived by patients and their families as a common and significant manifestation of the disease, strongly affecting quality of life and scholastic performance of children with NF1 [26]. 

Oncological and non-oncological manifestations contribute to the morbidity and mortality of NF1 patients. While resective surgery can be used to remove tumors, this is often challenging due to their location, size, and infiltrative nature. As for pharmaceutical treatments, few have been approved for treatment of NF1 oncogenic manifestations, one being selumetinib, targeting plexiform neurofibromas in children [27]. In addition, only a small number of investigations have suggested that treatment with stimulants may improve attention and cognitive function in individuals with NF1 who exhibit ADHD symptoms [23,28,29,30]. This most likely reflects still unexplored molecular complexity underlying these manifestations and demonstrates that much more research efforts must be directed towards understanding their pathogenesis for their successful amelioration. 

### 1.2. Variability of Symptoms

A major hallmark of NF1 is the high degree of variability and unpredictability of manifestations even between related individuals. This clinical heterogeneity can be attributed to several different factors, and these may contribute additively or differentially to NF1 presentations. 

A significant contributing factor is the large size of the gene spanning over 350 kb of genomic DNA, and containing 60 exons, some of which are alternatively spliced in a tissue-specific manner (Figure 1) [8,9,10]. A consequence of the large size of the gene is the plethora of missense, nonsense, and small deletion mutations identified in NF1 patients distributed over the different apparent domains of this large protein. Clearly, loss-of-function mutations, or those affecting the only well-characterized activity of the protein, the GAP-Related, GTPase-Activating Domain (GRD), are predicted to elevate Ras signaling. Nevertheless, mutations appear to be distributed nearly uniformly over other domains, which based on sequence homologies are an amino-terminal Cysteine- and Serine-Rich Domain (CSRD), a Tubulin-Binding Domain (TBD), a Sec14 Homologous Domain (SHD), a Pleckstrin Homologous Domain (PHD), and a Nuclear Localization Signal (NLS) in the carboxy-terminal domain (CTD) [31]. These domains are thought to mediate interactions with other proteins whose identity and role are not satisfactorily elucidated yet. Therefore, it is unclear how the majority of Nf1 mutations distributed over domains other than the GRD affect the localization and function of the protein. Since the precise contribution of the additional domains is not fully elucidated [31], we will refer to these mutations as apparent reduced-functionality alleles. However, it remains unclear how mutations outside the GRD domain might affect its Ras activity regulator role, or other still not well-established Nf1 functions. For example, whether, as predicted by the NLS, the protein actually enters the nucleus in vivo, the question of what its function(s) is therein, and whether they are perturbed by mutations and which ones.

The multitude of reduced functionality mutations is thought to be a significant contributor to the clinical heterogeneity and although a few thousand such mutations have been identified, the genotype-to-phenotype correlation remains poor [32,33,34,35]. In fact, some mutations have been linked solely or predominantly to cognitive deficits without tumors and other skin manifestations, while others primarily with the opposite presentation. This agrees with the working hypothesis that this variability is the result of mutations affecting distinct functional domains of the protein [36]. Additionally, second-hit events in the *NF1* gene leading to somatic mosaicism, genetic modifiers, epigenetic changes, developmental effects, and finally sex differences may contribute to the genotype/phenotype discord [37,38,39]. 

Another source of genotype/phenotype discord that could add to the phenotypic variability is the increased probability for somatic recombination due to the large size of the gene. Somatic recombination can lead to loss of heterozygosity (LOH), resulting in cells and tissues homozygous for mutations. This has been suggested to underlie the skin and tumor manifestations and result in cells homozygous for mutations or deletions, whereas others will remain heterozygous or even with two normal alleles [40,41]. Although this in principle can be a major contributor to phenotypic variability, it is unclear if such mechanisms may in fact underlie behavioral and cognitive deficits in the non-mitotically active neurons. 

Recent evidence suggested an alternative mechanism of reducing NF1 without apparent LOH or second-hit events based on the breakthrough discovery that the protein can form dimers in vitro [42,43]. However, it is presently unknown whether Nf1 is constitutively or conditionally dimeric, whether all isoforms can dimerize with each other, or whether dimerization is isoform-selective and if dimerization occurs only in particular tissues or developmental stages. Significantly, point mutations shown to disrupt dimerization in vitro destabilize not only the mutant protein, but also its wild-type counterpart encoded by the normal allele, apparently leading to its proteasome-dependent degradation [44]. The net result of such destabilizing mutations would be a significant reduction in both the variant and the wild-type protein in the heterozygous patients, potentially leading to exacerbation of pathology. Although it needs to be validated in an animal model in vivo, this evidence provides a putative mechanism to correlate the genotype to phenotype, at least for these destabilizing mutations. 

### 1.3. The Necessity of Animal Models

Progress towards understanding the pathologies and dysfunctions associated with NF1, including the genotype-to-phenotype discord, requires elucidation of the molecular and cellular mechanisms underlying them. Oncogenic manifestations can be modelled in cell culture and organoid systems emulating mutations in patients [45]. However, modeling NF1 cognitive and social deficits clearly requires animal models capable of such behavioral readouts. In addition, such animal models can also contribute towards a mechanistic understanding of the oncogenic and developmentally derived deficits associated with NF1. 

The genetic or RNA-interference-mediated attenuation of Nf1 in animal models should precipitate phenotypes similar or analogous to those in patients. An animal model may not present all NF1 manifestations, but this does not render it less disease-relevant. In fact, this divergence may reflect differential tissue-specific Nf1 functions among the various species used as models. For example, although mouse models of Nf1 develop most hallmark developmental and cognitive deficits, they do not present with apparent freckling or CALs [46]. These skin manifestations are present in the minipig model along with all other relevant pathologies in patients [47]. Ideally, an animal model should offer an expedient emulation of patient mutations of interest, but also be amenable to mechanistic studies on their effects on the function(s) of neurofibromin in multiple cell and tissue types. Without such mechanistic insights, effective ameliorative pharmacological regimes are unlikely, not to mention personalized therapeutic approaches strongly suggested by the genotype/phenotype discord. 

Nf1-mutant animals of various species, such as mice, zebrafish, and minipigs, have been generated to investigate the role of Nf1 and the consequences of its loss in various tissues, emulating many aspects of the human disease [48]. Each model along with work with organoids and cultured cells has contributed significantly towards our current understanding of signaling pathways engaging Nf1 and the association of the dysregulation of such pathways with growth and synaptic plasticity deficits manifested as impairments in learning and memory, social behaviors, and sleep patterns, among others [49,50,51,52,53,54,55,56]. These studies have provided significant insight regarding the molecular mechanisms and cellular circuitries affected by Nf1 loss including those that ostensibly contribute to learning disabilities observed in individuals with NF1. 

*Drosophila melanogaster*, the fruit fly, is a powerful genetic model that not only has been used to model human diseases, but is also of translational relevance [57]. *Drosophila* is highly cost-efficient and expedient, with a short generation time and many publicly available resources (http://flybase.org, accessed on 10 February 2024). As its genome encodes homologs or orthologs of 75% of human-disease-associated genes [58], it provides a highly versatile and expedient investigative tool for human neurological disorders. In addition, the well-established highly advanced and versatile transgenesis systems and spatiotemporal controls of transgene expression facilitate the introduction of human wild-type and mutant genes into the genome. These attributes along with the fast and efficient genome editing methods [59,60,61] amount to a premier model for the discovery and validation of genes and processes involved in physiology or pathology. This also pertains to aspects of behavior and cognition as *Drosophila* exhibits a rich neuroplasticity repertoire like attention, sleep, circadian activity, and multiple behavioral flexibility outputs including learning and memory [62], habituation [63,64,65,66], and social interactions [67], which are amenable to systematic, often medium–high throughput, analyses and quantification. Therefore, all methods and attributes of the system can be utilized to explore various aspects of behavioral pathology and cognitive deficits [68,69] pertinent to NF1, to facilitate elucidation of molecular mechanisms underlying them.

This review aims to summarize the contributions of *Drosophila* models of Neurofibromatosis 1 in advancing our understanding of the disorder and elucidation of its molecular, cellular, and behavioral aspects. Additionally, it seeks to identify knowledge gaps, emerging trends, and potential therapeutic strategies for this complex neurodevelopmental disorder.

## 2. Drosophila Models of Neurofibromatosis 1

### 2.1. The Drosophila Nf1 Gene and Protein (dNf1) Isoforms

Transcripts from the highly conserved Drosophila *dNf1* ortholog encompass 20 exons and give rise to five different transcripts due to alternative splicing mostly at the terminal noncoding 3′ exons, giving rise to alternative polyadenylation sites, while two mRNAs lack coding exon 14 (Figure 1) (http://flybase.org/reports/FBgn0015269, accessed on 10 February 2024). This is predicted to yield five protein isoforms of 2802, 2793, 2764, 2746, and 2734 amino acids, differing largely at their carboxy-termini and the presence or absence of residues encoded in exon 14. Of these, the 2802 (Nf1-PB) and the 2764 (Nf1-PD) isoforms have ostensibly been detected experimentally [70], although resolving all these ~280 kD isoforms by standard methods is rather difficult. Both experimentally detected isoforms are predicted to contain all recognized domains and share 54% identity and 68% similarity with their human counterpart. Interestingly, despite the 21 amino acid gap within the well-recognizable fly GRD domain relative to the human homolog (Figure 2), it retains its Ras-regulatory activity [70,71]. Additional domains readily recognizable in the fly protein are the CSRD and TBD, the lipophilic molecule-binding SHD/Sec14, and the phosphatidylinositol lipid-binding PH domain. A domain present in proteins involved in morphogenesis (MOR2-PAG1) is not readily apparent in the human protein, while conversely the nuclear localization signal of the human homolog is not recognizable in the fly protein and appears as a gap in the aligned sequences. Therefore, although homologous, there are domains that are not shared between the human and the fly Nf1 proteins (Figure 2).

The human (bottom line, blue) and Drosophila (Nf1-PB) (top line, red) neurofibromin proteins were aligned using both FlyBase (http://flybase.org/reports/FBgn0015269, accessed on 10 February 2024) and NCBI (https://blast.ncbi.nlm.nih.gov/Blast.cgi, accessed on 10 February 2024) protein BLAST tools. The fly protein is 54% identical and 68% similar to the human protein over its entire length. CSRD is the cysteine/serine-rich domain (Dm aa 582–928); TBD is the tubulin-binding domain (Dm aa 1133–1241); GRD is the GTPase-activating protein (GAP)-related domain (Dm aa 1242–1574); Sec14 is the bipartite lipid-binding module with a Sec14-like domain (Dm aa 1602–1750); PH is the pleckstrin homology domain (Dm aa 1759–1868); NLS is the nuclear localization signal (Hs aa 2555–2571, absent in Dm); and aa is amino acids.

The fly protein appears to be distributed primarily in the nervous system in all life stages [52,70,71], although the mRNA can be detected in many more cell types and tissues (http://flybase.org/reports/FBgn0015269, accessed on 10 February 2024). This suggests that as in vertebrates, dNf1 isoforms may be differentially distributed over many tissues and developmental stages, but the available antibodies to the Drosophila protein do not afford this level of resolution. Thus, to date, it remains unclear as to whether all predicted isoforms exist altogether and if so, whether they are differentially distributed in tissues and/or developmental stages. 

Loss of dNf1 results in organismal size reduction [52,71], attenuated associative learning and memory [51,52,72,73], deficits in circadian activities [55,74], behavioral inflexibility manifested as excessive grooming [54,75], aberrant sleep [56,76], and deficits in social behavior [77], summarized in Table 1. The behavioral defects may be consequent of impaired synaptic transmission [78,79], and possibly compromised metabolic homeostasis [80,81]. These deficits broadly resemble symptoms presented by human patients and therefore Drosophila can be employed to determine the molecular mechanisms within CNS neurons requiring dNf1 to mediate the behaviors compromised upon its loss. One significant difference with vertebrate models is that unlike humans and other vertebrate models except zebrafish [82], null mutations in dNf1 are homozygous-viable [71,83]. Unlike zebrafish that possess two distinct and redundant neurofibromin-encoding genes [82], there is a single *dNf1* gene in *Drosophila*. Because Ras signaling dysregulation is lethal in *Drosophila* [84], the fact that *dNf1*-null animals survive suggests that another, yet unidentified, GTPase Activating Domain-possessing protein may be recruited to regulate this developmentally important cascade.

Below, we will summarize the contributions of the *Drosophila* model towards understanding NF1 cognitive and behavioral pathological manifestations and progress towards ameliorative approaches for the consequent symptoms.

### 2.2. Modeling Νf1 Manifestations in Drosophila

#### 2.2.1. Associative Learning and Memory Defects

Children with NF1 present a constellation of behavioral deficits with learning disabilities and compromised executive function being among the most prevalent [23]. Verbal learning, reading, visuospatial skills, attention, emotionality, and social skills are significantly affected, in proportion to the severity of other symptoms [85]. Moreover, the prevalence of attention-deficit/hyperactivity disorder (ADHD) ostensibly affects cognitive and executive function variables and likely underlies the reduced scholastic performance and educational attainment of NF1 children [86]. Rational ameliorative strategies for these deficits require understanding the role of Nf1 in molecular mechanisms underlying learning and memory, which are compromised upon loss or altered functionality of the protein.

The learning disabilities associated with NF1 were one of the first disease manifestations modelled in *Drosophila* using an associative learning protocol. Among other currently available learning paradigms [62], the negatively reinforced associative learning [87] is the most robust and commonly used assay. Flies are conditioned by mild electric footshocks to selectively avoid a concurrently presented aversive odor from another equally aversive odor uncoupled to footshocks [87]. Loss of *dNf1* leads to significantly compromised associative learning [51,52,73] and memory [72], emulating human cognitive manifestations. Negatively reinforced associative learning and memory are known to engage the Mushroom Body Neurons (MBns), prominent bilaterally symmetrical structures consisting of ∼2000 intrinsic neurons (MBns) per brain hemisphere [88]. The MBn dendrites form the calyx, while their axonal projections form a characteristic tripartite structure [89]. 

Given the crucial role of the MBns in associative learning and memory, it was quite surprising that dNf1 is in fact not required within these neurons for associative learning. In contrast, dNf1 was shown to be required within GABAergic neurons, apparently presynaptic to the MBns, to curb their activity and permit associative learning within MBns [51]. Loss of dNf1 in these GABAergic neurons results in excess inhibitory neurotransmission to MBs and impaired associative learning and this effect is reversed if Ras signaling is reduced therein [51]. These findings are in agreement with results from *Nf1*^+/−^ mice, which also present elevated GABAergic neurotransmission reversible by Ras signaling inhibition, which rescues their cognitive deficits [49,50,90,91,92,93]. This also agrees with the enriched *Nf1* mRNA in mouse inhibitory neurons [94] and the interaction with the hyperpolarization-activated cyclic nucleotide gated K^+^/Na^+^ channel, HCN1 [92]. Properly controlled GABAergic signaling is cardinal to maintenance of the excitatory/inhibitory balance and stability within the central nervous system [95] and its perturbations have been linked to several neurodevelopmental disorders [96,97]. Therefore, both in flies and mice, Nf1 loss may not impair learning per se, but rather results in excess GABAergic signaling to MBns or the hippocampus to ostensibly inhibit learning. Despite early reports to the contrary [73], cAMP elevation did not reverse the learning deficit in flies [51].

A central question stemming from these results is how does Nf1 loss result in excess GABAergic signaling? Is it by dysregulated GABA release or excess GABA production resulting in excess tonic or even evoked release? In heterozygous null mice, Nf1 reduction results in enhanced ERK signals, which increase the pool of phosphorylated synapsin I, a condition promoting synaptic vesicle fusion and release [50]. Whether this affects GABAergic neurons specifically and why remained unclear. However, two recent publications suggest that dysregulated synaptic release may not affect only GABAergic neurons. Homozygous *dNf1*-mutant Drosophila larvae present tactile hyperexcitability and excess spontaneous cholinergic neurotransmission [78]. As for excess GABA release, the deficit was reversible by reducing Ras signaling in the cholinergic neurons or restoring a wild-type copy of dNf1 therein [78]. It is currently unknown whether excess GABA in adult CNS neurons and acetylcholine release in the larval ventral nerve chord is consequent of dNf1-dependent synapsin phosphorylation as in mice [49,98], or whether it involves other synaptic proteins engaged in synaptic vesicle fusion and neurotransmitter release. Nevertheless, unregulated synaptic activity in multiple neuronal subsets releasing different neurotransmitters may be a hallmark of Nf1 loss and could be Ras-hyperactivation-dependent or not. If so, what these neurons are and how their hypothetical dysregulation of neurotransmitter release relates to behavioral deficits in patients remains to be determined. The genetic facility and versatility of *Drosophila* may provide answers to this question, as elucidating the mechanisms of synaptic dysfunction in NF1 holds promise towards therapeutic strategies for the associated cognitive impairments. 

A potential therapeutic strategy emerged from work in *Drosophila*, which initially revealed the functional interaction of dNf1 with the Receptor Tyrosine Kinase (RTK) and Anaplastic Lymphoma Kinase (dAlk). This RTK is involved in cutaneous large-cell lymphoma and lung adenocarcinoma in humans [99]. dAlk and dNf1 are broadly colocalized in the larval and adult *Drosophila* CNS and reducing the dose of the RTK or inhibiting its activity with the FDA-approved drug TAE684 ameliorated the learning deficiency of *dNf1*-null flies [51,52]. Alk inhibition as an ameliorative treatment was also shown to be effective in *Nf1*^+/−^ mice [100]. Interestingly, unlike prior suggestions [83,101], activated PKA within these GABAergic neurons did not ameliorate the learning deficit. However, a Ras-dependent mechanism in MBns might modulate local cAMP levels in response to increased GABAergic stimulation, providing a plausible explanation of the suggested connection between cAMP levels and Nf1 function [51]. 

Since Alk signals through Ras to activate MAPK [52,102], these results indicate that excess signaling via this cascade underlies the learning deficits. In fact, the attenuation of dAlk levels within GABAergic neurons ameliorated the learning deficits of *dNf1* homozygotes [51], suggesting that the dAlk-dependent hyperactivation of Ras signaling upon dNf1 loss in these neurons results in excessive GABA release. It is unclear at the moment whether the physiological role of these neurons is to reversibly suppress learning as has been demonstrated for other GABAergic neurons impinging upon the MBns [103,104]. If so, Alk activation might drive this inhibition in a Ras-dependent manner and upon dNf1 loss, and this gating activity becomes constitutively inhibitory to learning. This putative pathway appears to be conserved since the Alk inhibitor also reverses the learning deficits in *Nf1*^+/−^ mice [100] and suggests that Alk may gate through regulated Ras activity hippocampus-dependent learning in mice and possibly humans. In humans, ALK is in fact expressed in NF1 tumors [105], supporting the potential therapeutic value of Alk inhibitors for NF1 cognitive dysfunction and underscoring the importance of regulated Ras signaling in learning and memory. In contrast, clinical trials that utilized statins to inhibit Ras signaling and improve cognitive outcomes in NF1 patients have demonstrated limited ameliorative efficacy [106,107,108,109,110,111,112]. The specific reasons for the disappointing results of these trials remain unclear, but several potential factors could contribute to this restricted effectiveness, including the complexities of Ras signaling pathways, potential statin off-target effects, the multifaceted nature of cognitive impairments in NF1, or the need for more targeted and personalized therapeutic approaches.

#### 2.2.2. Autism Spectrum Disorder Manifestations

Evidence suggests that 40% to 56% of individuals with NF1 demonstrate behaviors that broadly resemble those that characterize autism spectrum disorder (ASD) [21,113]. Compared to the typically highly heterogeneous manifestations in ASD individuals, NF1 children with NF1 present mostly pronounced social and communicative impairments and less of repetitive, inflexible behaviors. Because to date the genetic contributions to ASD are not well understood, the biological basis of this disabling comorbidity in NF1 is unclear. ASD is characterized by atypical sensory processing, which is distinguished from the neurotypical responses in how the brain interprets sensory information and reacts to sensory inputs (sight, sound, touch, taste, and smell). Importantly with respect to phenotypes in *Drosophila*, altered reactivity to sensory stimuli is now included as part of the diagnostic criteria for ASD under DSM-V (American Psychiatric Association, 2013). In *Drosophila*, altered reactivity to sensory stimuli could be manifested either as increased sensory sensitivity, or as a propensity to engage in sensory-seeking activities or behaviors. A measure of repetitive, ostensibly inflexible behaviors in *Drosophila* is persistent grooming, even in the absence of obvious stimuli such as dust, droplets, or strong odors. Grooming behavior in Drosophila is tightly regulated by the CNS and is also controlled by the circadian clock [114]. 

As revealed by King et al. [75], dNf1 loss or pan-neuronal dNf1 knockdown result in a robust seven-time increase in spontaneous grooming behavior, reflecting elevated grooming frequency and duration. This result clearly draws parallels to sensory-seeking behaviors associated with ASD. This persistent behavior appears to be the result of dNf1 loss during the development of sensory circuits that drive the behavior, resulting in heightened grooming in adulthood [54]. While the precise molecular and cellular mechanisms remain to be fully elucidated, the Ras signaling pathway emerges again as a pivotal player in regulating grooming behavior, with the Nf1 Ras-GAP activity being crucial for maintaining the behavior within normal levels [54]. Hence, emulating the inhibitory function of Nf1 in Ras-GTPase signaling [49,115], treatment of dNf1-deficient flies with drugs that prevent Ras hyperactivation could ameliorate aberrant grooming behavior and by extension be useful towards similar symptoms in patients.

Significantly, the neuronal circuits orchestrating grooming behavior are composed of both excitatory and inhibitory neuronal subsets. Optogenetic stimulation of the excitatory neurons in this circuit induces antennal grooming, while the simultaneous activation of parallel sensory pathways triggers a grooming sequence in *Drosophila*, shedding light on the multifaceted nature of sensory processing in modulating grooming behavior [116,117,118]. Considering that excessive grooming upon dNf1 loss of function originates from excitatory cholinergic and β-adrenergic-like octopaminergic neurons, a plausible hypothesis emerges: dNf1 deficiency may result in system overexcitation, manifesting as excessive grooming behavior. 

Interestingly, these findings may be relevant to the approximately 60% of children with ASD, which present differences in tactile sensitivity [119]. A probable reason for this presentation was offered by measuring neuronal activity in the neuromuscular junction (NMJ) of larval *Drosophila* [120]. Loss of dNf1 results in synaptic overgrowth at the NMJ, suggesting that the protein normally restrains synaptic growth within physiological constraints, and this engages the cAMP pathway, rather than its GAP activity [102]. Importantly, synaptic transmission from these overgrown synapses is also increased, probably resulting in excess muscle activity. In fact, dNf1 deficiency was shown to lead to significantly compromised evoked neurotransmitter release, yet increased spontaneous activity in the NMJ of null mutants [78]. Increased spontaneous transmission is typically indicative of neuronal hyperexcitability [121] and is often manifested behaviorally as larval tactile hypersensitivity. The effect seems to be developmental and mediated by central cholinergic neurons. Accordingly, cortical hyperexcitability in ASD is often suggested to underlie behavioral hypersensitivity and inappropriate neural responses to stimuli [122]. 

Therefore, excessive excitatory firing, arising from the central locomotor circuitry, appears to underlie tactile hypersensitivity in *dNf1*-null-mutant larvae. Knocking down Ras proteins pan-neuronally ameliorates both the tactile hypersensitivity and the overexcitation/hyperexcitability of the cholinergic neurons, suggesting that both deficits result from the overactivation of the MAPK pathway [78]. Reduced BK-Ca^2+^ channel levels or activity are also involved in the tactile hypersensitivity, as pharmacological intervention with the channel activator BMS-204352 ameliorated the deficit [79]. Collectively, these results suggest that dNf1 restricts spontaneous release. Upon its loss, however, spontaneous release is increased at the expense of evoked transmission, as has also been reported for synaptotagmin mutant larvae [123,124]. Importantly, exactly how dNf1 regulates synaptic development and regulates spontaneous activity awaits resolution. This requires the identification of dNf1 effectors within the different neuronal types engaged and whether they are participants in the Ras or other signaling pathways therein. 

A growing body of evidence indicates that children with NF1 present more social problems than unaffected individuals. Although willing to engage with others, children with NF1 with or without ASD encounter various difficulties in their social–communicative life. These social dysfunctions include attenuated social perception and reduced emotionality, face recognition, friendships, and empathy [125], in addition to typical ADHD manifestations [126]. This suggests that sensory perception and processing, especially of socially relevant stimuli, may be altered in NF1 individuals. Consistent with the notion of altered sensory perception contributing to and perhaps underlying social behavior defects, dNf1 loss in *Drosophila* sensory neurons was shown to lead to sensory processing errors [77]. Chemosensory neurons in these flies allow perception of pheromonal cues processed by an apparently dedicated circuit in the CNS, which drives courtship towards females and inhibits courtship towards male conspecifics. dNf1 is required within these peripheral chemosensory neurons and its loss impairs proper sensory cue perception, resulting in inappropriate indiscriminate courtship [77]. This effect is not developmental, as the deficits in social interactions are also caused by hyperactive Ras signaling within these sensory neurons upon acute dNf1 loss. 

Therefore, social deficits in this *Drosophila* NF1 model are consequent of sensory neuron dysfunction. Interestingly, dysfunction in peripheral sensory neurons is thought to contribute to core behavioral features across a range of ASD models [127,128]. In fact, one influential model posits that sensory impairments give rise to errors in brain development, leading to behavioral deficits in adulthood [129]. Social and cognitive deficits can be mapped to primary dysfunction of somatosensory neurons rather than neurons within the CNS. Therefore, targeting sensory neurons to ameliorate social skill and social responsiveness deficits in NF1 patients in clinical trials is a possibility [126]. The facility of the *Drosophila* model can expedite identification of candidate pharmaceuticals, as male–male courtship assays can be automated to a medium throughput. 

#### 2.2.3. Sleep Fragmentation and Activity Deficits 

Several reports indicate a high occurrence of sleep disturbances in NF1 patients, especially children, implying that neurofibromin may regulate sleep [130,131,132]. More specifically, deficits in sleep initiation and maintenance have been mostly described in children with Neurofibromatosis type 1, thus far [18,19,20]. Although sleep disturbance typically is not a major complaint of affected individuals, parents often report that children with NF1 complain of overwhelming tiredness, exhaustion, and lack of energy [132], which may be consequent of sleep fragmentation. 

Although the exact purpose of sleep is unknown, it is conserved in all animal species that have been examined thus far [133]. *Drosophila* has been at the forefront of sleep and circadian genetics, and several studies have yielded important insights into the molecular and neural circuits that drive these behavioral processes. These flies exhibit an ∼24 h activity cycle independent of external cues, such as temperature and light. The circadian clock controls various rhythmic behaviors, including daily changes in locomotor activity. Locomotor activity refers to patterns of movement and rest, which is a complex trait regulated by many interacting loci with small effects that are sensitive to environmental factors [134]. Each activity cycle is characterized by distinct periods of sleep, defined as any period of immobility lasting for more than 5 min, and shares features of sleep in humans [135,136]. Importantly, *dNf1*-null flies present significantly disrupted sleep [56,76], as also presented by *Nf1*^+/−^ mice [137]. Both *dNf1*-null flies and *Nf1*^+/−^ mice present sexual dimorphism with substantially shorter sleep and sleep fragmentation in males compared to females [56,137], suggesting sex-differential sleep regulation. Interestingly, dNf1 functionally interacts with dAlk in sleep regulation [56,76], as it does in the context of learning [51,52]. Accordingly, genetic inhibition of Alk rescued learning and sleep behaviors in *Nf1*^+/−^-mutant mice [100,138]. These results strongly reinforce the rationale for exploring the therapeutic potential of Alk inhibition [139] in reversing both cognitive- and sleep-related symptoms in individuals with NF1.

Adult dNf1 expression is required for normal sleep in *Drosophila* across its life span [140], but sleep duration and depth are apparently developmental in origin [76]. Adult flies lacking dNf1 apparently fail to enter deep sleep, presenting sleep fragmentation, reduced arousal threshold, and loss of sleep-dependent reductions in the metabolic rate [141]. These results suggest that dNf1 functions in distinct circuits regulating sleep duration and regulation of the metabolic rate. These metabolic and sleep phenotypes of the mutants are phenocopied in flies where dNf1 is specifically knocked down in GABA_A_ receptor/Rdl-expressing neurons, thus raising the possibility that sleep dysregulation is due to altered GABA signaling. This notion is supported by previous findings that dNf1 knockdown in GABA_A_-expressing neurons results in shortened sleep bouts and reduced sleep duration [140] and the dysregulated GABA signaling to the MBs underlying learning deficits [51]. These findings raise the possibility that the Ras-signaling-dependent GABAergic modulation of sleep quality might affect the metabolic rate [141]. However, due to the pleiotropic effects of dNf1 loss on both behavior and physiology, establishing a direct connection between sleep depth disturbances and the metabolic rate poses a challenge. 

Loss of dNf1 also disrupts circadian locomotor activity. Null mutants are arrhythmic [55] and pan-neuronal knockdown of dNf1 recapitulates the phenotype [142]. However, because in dNf1 mutants, the key circadian clock proteins, period (PER) and timeless (TIM), cycle normally in central clock cells, loss of dNf1 does not affect the function of the clock itself, but rather its downstream outputs. In fact, the circadian phenotype is rescued by loss-of-function mutations in the Ras/MAPK pathway, establishing the link between Ras/MAPK signaling and Nf1 function in circadian rhythmicity [55]. dNf1 regulates circadian rhythms by modulating cAMP levels in clock neurons, which affects the expression of clock genes and the activity of downstream targets [74]. Therefore, the dysregulation of cAMP levels in this circuit likely underlies the circadian activity changes upon dNf1 loss and may also affect sleep-related presentations in NF1 patients.

In addition to disturbed sleep, knockdown and null mutations of dNf1 also result in significant night-time hyperactivity [75,140,142], effectively phenocopying the acute activation of dopaminergic neurons [143]. These dNf1-loss-associated, light-dependent hyperactivity and sleep signatures implicate disrupted dopamine signaling [142]. Because of the consequent reduction in cAMP levels, aberrant dopaminergic signaling has also been implicated in attention and cognitive deficits in mouse NF1 models [144,145,146]. Historically, children with NF1 who experience attention deficits have been prescribed stimulants, such as methylphenidate (MPH), which increases extracellular dopamine [147,148] and improves their performance [29,30]. Mutations in *Nf1* increase ADHD risk by approximately eightfold, and ADHD is diagnosed in nearly every second child with NF1. Remarkably, methylphenidate treatment managed to normalize Nf1-related hyperactivity and sleep phenotypes in *Drosophila* as well. Thus, the night hyperactivity and sleep defects of dNf1 mutants represent an ADHD-relevant endophenotype in *Drosophila* that can be rescued by MPH [142]. This is another confirmation of the translational relevance of the *Drosophila* NF1 model and its utility in the discovery and experimental validation of ameliorative pharmaceuticals for the plethora of symptoms associated with the condition. 

#### 2.2.4. Stature and Metabolic Homeostasis Dysregulation

Children with NF1 are born smaller in size and develop shorter in height than average adults [149,150]. This indicates potential disruptions in growth and developmental processes, hypothesized to be linked to metabolic regulation. In fact, studies in both humans and animal models have revealed evidence indicating that Nf1 apparently plays a role in regulating metabolism, although the precise mechanisms are not yet fully understood. Observations indicate that affected individuals tend to present with deviations in their metabolic profiles, which may contribute to their smaller size and stature. Additional studies have revealed a lower prevalence of diabetes mellitus (mainly type 2) among NF1 patients, suggesting a complex interplay with signals and molecular pathways involved in glucose homeostasis and insulin signaling [151]. In fact, reduced thalamic glucose metabolism could be related to cognitive impairments [152].

*Drosophila dNf1* mutants are significantly smaller in pupal- and adult-size compared to control animals of the same genetic background [71]. This was shown to be consequent of smaller cell size rather than fewer cell numbers [52] and to result from Ras/MAPK signaling overactivation [52,71]. Significantly, pharmaceutical inhibition of dAlk fully rescued the size defect, verifying that this RTK is upstream of the dNf1-regulated Ras activation relevant to size determination [52]. However, the upregulation of cAMP/PKA signaling within neuroendocrine cells secreting insulin-like peptides also reversed the growth deficit, suggesting interactions with the dAlk-mediated dNf1-regulated Ras signaling [105,153]. By analogy, these results suggest that the reduced body size of patients is indeed likely due to perturbed glucose homeostasis during embryonic, infant, and adolescent development. The power and advanced tools of *Drosophila* genetics can be used to address this hypothesis and provide further understanding of the interaction between the dAlk-initiated signals and cAMP signaling and may even provide novel ameliorative pharmaceutical targets. Growth defects may also underlie the skeletal malformations and deficits presented by many NF1 patients [17,154,155,156]. Although an invertebrate, findings from investigations on growth control in *Drosophila* may inform genetic or pharmacological screens to identify ameliorative approaches to bone density and skeletal defects in patients. 

**Table 1 cells-13-00721-t001:** Cognitive and behavioral phenotypes in Nf1 Drosophila models.

Deficient Phenotype	Mutation	Implicated Pathway	Drug Treatment	Refs.
Olfactory associative learning and memory	null mutants; RNAi KD in OK72/GABA neurons	cAMP Alk/Ras1	TAE684	[51,52,72,73,101]
Circadian rhythms	null mutants; RNAi KD in MB neurons	Ras cAMP		[55,56,74]
Locomotor activity	null mutants; pan-neuronal RNAi KD		MPH	[75,142]
Sleep	null mutants; pan-neuronal RNAi KD; RNAi KD in GABA_A_R-expressing neurons (Rdl-Gal4); RNAi KD in MB neurons	Alk (Ras) cAMP	MPH	[56,76,140,141,142]
Grooming behavior	null mutants; RNAi KD pan-neuronal, in VNS and in ChAT- and oct-tyrR-expressing neurons	Ras		[54,75]
Social behavior	null mutants; RNAi KD in Fru^+^ sensory neurons; RNAi KD in Ppk23^+^ sensory neurons	Ras		[77]
Synaptic transmission	null mutants; pan-neuronal RNAi KD (pre-synaptic); RNAi KD in cholinergic neurons (ChaT-Gal4)	Ras BKCa channel cAMP	simvastatin BMS-04352	[78,79,83,102]
Metabolic homeostasis	null mutants; pan-neuronal RNAi KD; RNAi KD in VNS, Oct-TyrR, and PCB neurons	cAMP Ras		[80,81,140]
Growth	null mutants	Alk (Ras) cAMP	TAE684	[6,52,70,71,102,105,153]

All deficient phenotypes discussed in the text are listed, including the genetic makeup of the Drosophila model being studied, the implicated signaling pathways, and the pharmacological treatment used. Grey shading is used when more than one signaling pathway are implicated in the relevant phenotype. KD, Knockdown; MB, Mushroom Body; VNS, Ventral Nervous System.

Furthermore, Botero et al. in 2021 also revealed that neurofibromin governs metabolic homeostasis in *Drosophila* through a specific neuronal circuit. Loss of dNf1 led to widespread metabolic disruptions, including a markedly higher, Ras-activity-dependent metabolic rate in these flies, as has also been described in patients [157]. dNf1 loss triggered a homeostatic increase in feeding, along with alterations in lipid stores and turnover kinetics. Notably, the elevation in the metabolic rate was linked to a small set of neurons in the fly ventral nervous system, indicating the importance of elucidating potential connections between neuronal activity and metabolic processes in NF1 [80]. These observations imply that metabolic dysregulation might play a crucial role in the pathogenesis of NF1, contributing to the complex clinical manifestations associated with the disorder and providing a foundation for further exploration of the neuronal role of neurofibromin in maintaining metabolic homeostasis.

These metabolic consequences upon dNf1 loss may be reflected in the life span decrease and increased susceptibility to heat or oxidative stress [81]. The sensitivity to oxidative stress is apparently linked to reduced mitochondrial respiration and a greater production of reactive oxygen species (ROS) in *dNf1*-null mutants [81]. Interestingly, these phenotypes were also reversed through pharmacological or genetic interventions enhancing cAMP/PKA signaling. Moreover, the introduction of catalytic antioxidants to counteract ROS production restored a normal lifespan in these *dNf1*-null mutants [81]. These findings highlight the critical role of dNf1 in regulating mitochondrial function and ROS production, with potential implications in the development of therapeutic strategies. 

## 3. Perspectives and Concluding Remarks

Modeling NF1 pathological manifestations in Drosophila and using the advanced genetic toolbox of the system to map the dysfunctions to specific neurons of the fly CNS constitutes a significant advance in understanding the pathogenic mechanisms of the condition and potential ameliorative pharmacological approaches. 

One significant conclusion is that dNf1 is involved in regulated synaptic activity, at least from cholinergic [78], GABAergic [51], ostensibly dopaminergic [142], and neuroendocrine neurons [105]. This predicts that the protein must interact with proteins essential for regulated, especially evoked, neurotransmission, such as synapsins, synaptotagmins, and channels, among others. If verified, this information will not only elucidate the synaptic role of Nf1 but could lead to specific ameliorative pharmaceuticals [79]. Importantly, the FDA-approved Alk inhibitors, first described as ameliorative agents for the size and learning deficits in flies [52] and then in mice [139], may actually ameliorate sleep fragmentation, ASD- and ADHD-like manifestations, or excessive baseline cholinergic synaptic activity. However, given the variability of NF1 manifestations, it is equally likely that additional pharmaceuticals alone or in combination with verified effective ones may yield better ameliorative outcomes for distinct pathological manifestations. 

Of equal importance is the emerging picture that in addition to Ras signaling, Nf1 appears to regulate cAMP levels in a cell-type-specific manner. Although one such molecular cascade has been reported [158] for a particular vertebrate cell type, additional such mechanisms have been suggested largely by in vitro work in Drosophila [159], but have not been verified in other systems or in vivo. Because the Nf1-mediated modulation of cAMP levels appears to be cell-type-specific and may in fact underlie some, but not all pathologies, it could be selectively targeted pharmacologically. 

The recent findings that dNf1 in particular neurons regulates metabolism in *Drosophila*, in accord with the increased resting metabolism reported for NF1 patients [157], will most likely lead to determination of the underlying molecular mechanisms and circuitry. As metabolic perturbations most likely affect neuronal function including cognition, these findings open a new investigative approach towards understanding the NF1 pathobiology, potentially relevant to tumorigenesis as well [160], an aspect of the disease where Drosophila has not contributed significantly, to date at least. 

Despite the significant advances in our understanding of the complexity of mechanisms engaging Nf1 and the varied manifestations of its loss, it is critical that with few exceptions, protein null alleles are used. This approach provided significant answers as to what processes and molecular pathways Nf1 engages. However, a few thousand point mutations and small deletions have been identified in the human *Nf1* gene [33,34,35], affecting different domains of the protein, and have not been studied in animal models with the exception of a few recently reported in mice [45]. Given the diversity of NF1 symptoms and the number of distinct mutations in patients, addressing the question of whether distinct mutations result in distinct disease manifestations is of cardinal importance to rational ameliorative strategies. In addition, point mutations or small in-frame deletions may result in dominant negative alleles rather than loss of function, and this may affect all Nf1 functions that are domain-specific. In fact, a couple of destabilizing point mutations, not only for the protein that bears them, but for the co-expressed wild-type allele as well, have been reported recently [44]. It is unknown if additional such point mutations in other Nf1 domains result in analogous destabilization of the mutant and the wild-type proteins globally, or in a tissue-specific manner. The facility and expedience of generating CrispR point mutations coupled to the broad and varied repertoire of phenotypic assessments detailed above make Drosophila the premier system to address these questions in the first instance. Expedient and systematic functionalization and thorough assessment of human mutations that result in particular manifestations (i.e., largely cognitive deficits versus oncogenic) is essential towards development of rational and personalized ameliorative or even therapeutic strategies. 

## Figures and Tables

**Figure 1 cells-13-00721-f001:**
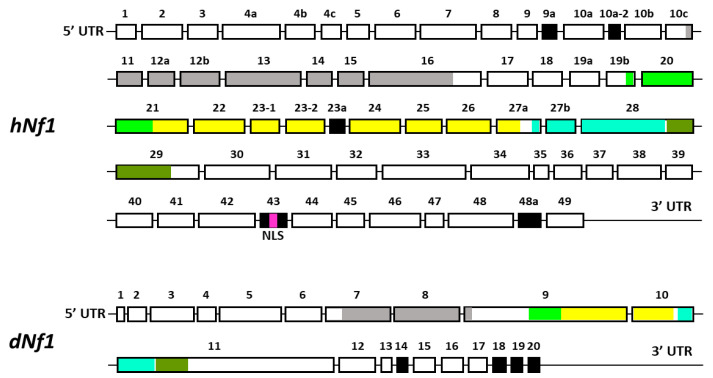
A schematic representation of the exons of the full-length transcripts of human and *Drosophila* Nf1. The alternatively spliced exons are indicated in black. Exons encoding different known domains of Nf1 are indicated in different colors (same color code as in Figure 2)—Gray: CSRD, cysteine/serine-rich domain; Light green: TBD, tubulin-binding domain; Yellow: GRD, GTPase-activating protein (GAP)-related domain; Light blue: Sec14, bipartite lipid-binding module with a Sec14-like domain; Olive green: PH, pleckstrin homology domain; Purple: NLS, nuclear localization signal.

**Figure 2 cells-13-00721-f002:**
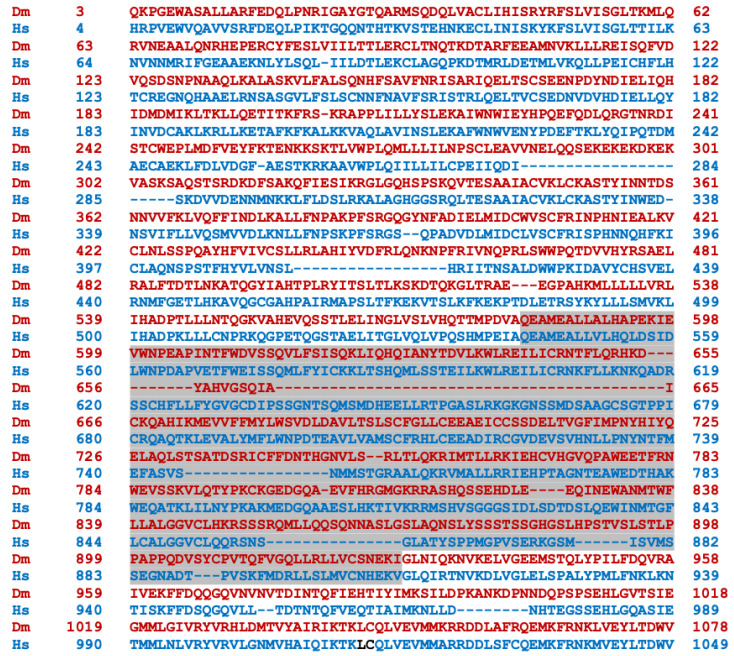
Alignment of the human and Drosophila neurofibromin proteins. The human (bottom line, blue) and Drosophila (Nf1-PB) (top line, red) neurofibromin proteins were aligned using both FlyBase (http://flybase.org/reports/FBgn0015269) and NCBI (https://blast.ncbi.nlm.nih.gov/Blast.cgi) protein BLAST tools. The fly protein is 54% identical and 68% similar to the human protein over its entire length. CSRD, cysteine/serine-rich domain (Dm aa 582-928); TBD, tubulin-binding domain (Dm aa 1133-1241); GRD, GTPase-activating protein (GAP)-related domain (Dm aa 1242-1574); Sec14, bipartite lipid-binding module with a Sec14-like domain (Dm aa 1602-1750); PH, pleckstrin homology domain (Dm aa 1759-1868); NLS, nuclear localization signal (Hs aa 2555-2571, absent in Dm); aa, amino acids.

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
