# Peer review of "Drosophila Contributions towards Understanding Neurofibromatosis 1"

_cells, 2024, doi:10.3390/cells13080721_

Round 1
Reviewer 1 Report
Comments and Suggestions for Authors
The neurofibromatosis type 1 (NF1) is caused by mutations in the neurofibromin gene (Nf1). This protein is expressed in a wide variety of neurons and glia in both the central and peripheral nervous systems. NF1 in humans has a broad spectrum of manifestations ranging from the appearance of neurofibromas (benign tumors generated by neurons of the peripheral nervous system) to autism, intellectual deficits, attention problems, sleep problems, and even growth deficits. One of the major challenges in seeking remedies that cure or at least improve this wide symptomatology is the great variety of mutations that can lead to NF1, making it necessary to have non-human models to test possible therapies.
This review extensively lists the cases in which the different symptoms of NF1 are emulated by their corresponding model in Drosophila, and suggests this model as very convenient for studying drugs, since thanks to the new CRISPR gene editing techniques, it is relatively easy to create fly lines with each of the mutations found in human patients. Furthermore, most neurological symptoms that may be difficult to measure in humans are usually manifested as more primary behaviors in Drosophila and therefore more easily quantifiable.
In my opinion, this review is a useful tool not only for researchers working on NF1 but also opens the expectation that many complex diseases can be modeled in Drosophila, where drug trials are not only faster but also much more economically and ethically convenient.
Comments on the Quality of English LanguageEnglish is good.
Author Response
We thank the reviewer for the kind comments.
Reviewer 2 Report
Comments and Suggestions for Authors
In this review manuscript titled “Drosophila contributions towards understanding neurofibromatosis 1,” Atsoniou et al. review what is known about NF1 and contributions towards our understanding of NF1 from the Drosophila model. After discussing NF1, symptoms, and the necessity of animal models to study NF1, the authors present a detailed and comprehensive analysis of the Drosophila Nf1 literature. Overall, this is a nicely written review that presents a good overview of the literature and clearly discusses the links between the fly and vertebrate studies. It does an excellent job of pointing out gaps in our understanding, as well as formulating testable hypotheses to stimulate further research. The narrative style and clarity of identification of gaps in the literature and links between flies and human patients is rather elegant. It also really has a strength in diligently pointing out the similarities between the Drosophila phenotype and patient symptoms, as well as illustrating how the fly can be used to learn more about the human disease. It is a useful contribution to the literature that will be of interest to the neuroscientists as well as both vertebrate and Drosophila biologists who study NF1. I have the following suggestions to improve the manuscript.
It would be useful to have a Figure illustrating the human Nf1 gene, as it is a complex locus, generates multiple isoforms, and has multiple functional domains. Visualizing this would be useful regarding the text on pages 2-3.
There are numerous grammar and punctuation errors that should be fixed. For the most part, they do not impact the clarity and understanding of the manuscript, and the manuscript has a clear and defined organization that makes it easy to follow. The formatting of section and subsection titles should be consistent.
Lines 137-155: I think these paragraphs are meant to introduce requirements of animal models of NF1, and to provide a motivation for generating animal models. However, this is not clear, and the paragraphs could benefit from rewording.
Line 277, 184, etc. Drosophila should be in italics, Drosophila, as it is a genus name.
Table 1: The relevant references should also be included in the table, perhaps by listing the relevant study reference by the deficient phenotype in the first column, or creating a fifth column in the table.
Section 2 (Lines 195-215): It would be useful here to have a figure illustrating the structure of the dNf1 locus and features, in addition to the link to Flybase, so that the reader can visually see this in the manuscript without having to link to a different website. Also, the Flybase release that this gene model is based on should be cited, in addition to the link to the gene report page (as gene models and FBgn numbers for complex loci frequently change with annotation updates).
Line 392 – reword “most probable” to “a probable”, as while Drosophila results can offer insight into a possible mechanism, it is hard to claim they demonstrate the most probable reason for a human neurological phenotype.
Lines 478-482 – it is unclear how reduced sleep and learning deficits would affect gut homeostasis. These seem like two completely different topics.
Lines 542-544 – This sentence is confusing, as it introduces new information, tries to identify flies as a useful model to screen therapies and concludes the paragraph. It could be broken into at least two sentences, the second of which could communicate that although Drosophila is an invertebrate, it is a useful and relevant model to screen and identify ameliorative approaches to bone density and skeletal defects in patients.
For Figure 2, where are the data plots from? Do these need citations, if they come from actual studies? Or are they AI generated in BioRender?
Comments on the Quality of English LanguageThe manuscript is well organized and structured, and the logic is easy to understand. There are numerous grammar and punctuation errors that should be addressed, too many to list individually.
Author Response
Overall, this is a nicely written review that presents a good overview of the literature and clearly discusses the links between the fly and vertebrate studies. It does an excellent job of pointing out gaps in our understanding, as well as formulating testable hypotheses to stimulate further research. The narrative style and clarity of identification of gaps in the literature and links between flies and human patients is rather elegant. It also really has a strength in diligently pointing out the similarities between the Drosophila phenotype and patient symptoms, as well as illustrating how the fly can be used to learn more about the human disease. It is a useful contribution to the literature that will be of interest to the neuroscientists as well as both vertebrate and Drosophila biologists who study NF1.
We thank the reviewer for the kind comments.
I have the following suggestions to improve the manuscript.
It would be useful to have a Figure illustrating the human Nf1 gene, as it is a complex locus, generates multiple isoforms, and has multiple functional domains. Visualizing this would be useful regarding the text on pages 2-3.
We have generated this now Figure 1 where the genomic organization of the human and fly genes are both included for ready comparison. Thanks for suggesting this, we also find it very useful.
There are numerous grammar and punctuation errors that should be fixed. For the most part, they do not impact the clarity and understanding of the manuscript, and the manuscript has a clear and defined organization that makes it easy to follow. The formatting of section and subsection titles should be consistent.
We believe we have fixed all such mistakes and formatting issues.
Lines 137-155: I think these paragraphs are meant to introduce requirements of animal models of NF1, and to provide a motivation for generating animal models. However, this is not clear, and the paragraphs could benefit from rewording.
This section now reads:
Progress towards understanding the pathologies and dysfunctions associated with NF1, including the genotype to phenotype discord, requires elucidation of the molecular and cellular mechanisms underlying them. Oncogenic manifestations can be modelled in cell culture and organoid systems emulating mutations in patients [45]. However, modeling NF1 cognitive and social deficits, clearly requires animal models capable of such behavioral readouts. In addition, such animal models can also contribute towards a mechanistic understanding of the oncogenic and developmentally derived deficits associated with NF1.
Genetic or RNA-interference-mediated attenuation of Nf1 in animal models should precipitate phenotypes similar or analogous to those in patients. An animal model may not present all NF1 manifestations, but this does not render it less disease relevant. In fact, this divergence may reflect differential tissue-specific Nf1 functions among the various species used as models.
We hope that is more along the lines the reviewer was thinking. Thanks for pointing this out.
Line 277, 184, etc. Drosophila should be in italics, Drosophila, as it is a genus name.
Thanks for pointing out this oversight!
Table 1: The relevant references should also be included in the table, perhaps by listing the relevant study reference by the deficient phenotype in the first column, or creating a fifth column in the table.
Done
Section 2 (Lines 195-215): It would be useful here to have a figure illustrating the structure of the dNf1 locus and features, in addition to the link to Flybase, so that the reader can visually see this in the manuscript without having to link to a different website. Also, the Flybase release that this gene model is based on should be cited, in addition to the link to the gene report page (as gene models and FBgn numbers for complex loci frequently change with annotation updates).
Done, see Figure 1
Line 392 – reword “most probable” to “a probable”, as while Drosophila results can offer insight into a possible mechanism, it is hard to claim they demonstrate the most probable reason for a human neurological phenotype.
We agree with the reviewer. Thanks for pointing this out!
Lines 478-482 – it is unclear how reduced sleep and learning deficits would affect gut homeostasis. These seem like two completely different topics.
Agreed it is bit tenuous, se we removed this part.
Lines 542-544 – This sentence is confusing, as it introduces new information, tries to identify flies as a useful model to screen therapies and concludes the paragraph. It could be broken into at least two sentences, the second of which could communicate that although Drosophila is an invertebrate, it is a useful and relevant model to screen and identify ameliorative approaches to bone density and skeletal defects in patients.
We believe we have fixed this. This part now reads:
Growth defects may also underlie the skeletal malformations and deficits presented by many NF1 patients [17, 154-156]. Although an invertebrate, findings from investigations on growth control in Drosophila may inform genetic or pharmacological screens to identify ameliorative approaches to bone density and skeletal defects in patients.
For Figure 2, where are the data plots from? Do these need citations, if they come from actual studies? Or are they AI generated in BioRender?
These are all BioRender-generated and the Figure has now moved to be the graphical abstract.
Reviewer 3 Report
Comments and Suggestions for Authors The authors provide a comprehensive summary of the clinical heterogeneity manifestations of Neurofibromatosis 1 (NF1) disorder in humans, as well as the contributions of Drosophila in modeling these NF1 manifestations. This manuscript offers valuable insights into utilizing Drosophila as an organism model to investigate the underlying molecular mechanisms of NF1, which could pave the way for future research in this field. Given its relevance to readers interested in NF1, I recommend this review for publication in Cells. Before publishing in Cells, I suggest the following minor revisions: 1. In Figure 1, to clearly depict the conservation information of NF1 between humans and Drosophila, using specific conservation analysis software like CLUSTALW multiple alignment instead of protein blast would be beneficial to reveal this information. 2. At line 252, the "of" in "towards ameliorative approaches of the consequent symptoms" changes to "to" for better clarity. 3. At line 615, "In addition, point mutation " should be " In addition, point mutations." 4. At line 626, "is" should be changed to "are" in "Expedient and systematic functionalization and thorough assessment of human mutations …are essential towards development of rational and personalized."
Author Response

(The authors gave the same response as above.)

Round 2
Reviewer 2 Report
Comments and Suggestions for Authors
The authors have addressed all of my concerns, and both the text edits and new figure have improved the manuscript.